# Control of quantum electrodynamical processes by shaping electron wavepackets

Liang Jie Wong [1✉], Nicholas Rivera [2], Chitraang Murdia [2], Thomas Christensen [2], John D. Joannopoulos[2], Marin Soljačić[2] & Ido Kaminer [3✉]

Fundamental quantum electrodynamical (QED) processes, such as spontaneous emission and electron-photon scattering, encompass phenomena that underlie much of modern science and technology. Conventionally, calculations in QED and other field theories treat incoming particles as single-momentum states, omitting the possibility that coherent superposition states, i.e., shaped wavepackets, can alter fundamental scattering processes. Here, we show that free electron waveshaping can be used to design interferences between two or more pathways in a QED process, enabling precise control over the rate of that process. As an example, we show that free electron waveshaping modifies both spatial and spectral characteristics of bremsstrahlung emission, leading for instance to enhancements in directionality and monochromaticity. The ability to tailor general QED processes opens up additional avenues of control in phenomena ranging from optical excitation (e.g., plasmon and phonon emission) in electron microscopy to free electron lasing in the quantum regime.

[1] School of Electrical and Electronic Engineering, Nanyang Technological University, Singapore, Singapore. [2] Department of Physics, Massachusetts Institute of Technology, Cambridge, MA, USA. [3] Department of Electrical Engineering, Technion, Haifa, Israel. ✉email: liangjie.wong@ntu.edu.sg; kaminer@technion.ac.il

Free-electron-driven technologies lie at the heart of modern science and engineering, from X-ray tubes used in medical imaging, industrial quality inspection, and security scanning, to electron microscopes that can capture fundamental phenomena with sub-angstrom[1,2] and sub-picosecond resolution[3,4]. The useful range of electron kinetic energies runs the gamut from non-relativistic energies, as low as 50 eV in applications like coherent low-energy electron microscopy[5–10], to ultra-relativistic energies, as high as several GeV in X-ray free electron laser facilities[11,12]. Broader applications of free electron sources include electron beam lithography[13,14], atom-by-atom matter assembly[15], nanoscale radiation sources[16–36], and electron microscopy[37–47].

The wide range of free-electron-based applications highlights the importance of developing effective electron waveshaping techniques, which would enable an even larger design-space in tailoring free-electron-based processes. Free electrons are readily manipulated through electron–light and electron–matter interactions, as manifested by phenomena such as the Kapitza–Dirac effect[48–51] and electron double-slit interference[52,53]. In particular, the structuring of an electron's wavefunction via interference has been experimentally demonstrated[53]. Just as optical waveshaping has uncovered a wealth of electromagnetic phenomena[54–59], so electron waveshaping promises to be rich in exciting electron beam physics and applications. A host of methods has arisen for the design of electron wavepackets, leveraging a variety of mechanisms—including static fields[60–65], radio-frequency cavities[66–71], laser pulses[72–83], and material structures[84,85]—to shape the spatiotemporal profile of an electron pulse, achieving temporal shaping down to the attosecond timescale. Breakthroughs in manipulating the phase structure of electron wavepackets[86,87] have led to further control over properties such as orbital angular momentum (OAM)[88–90], spin angular momentum[91,92], and propagation trajectory[93,94]. These structured electron beams can be generated through a variety of means including amplitude and phase holograms[95–99], nanoscale magnetic needles[100], and electron–photon interactions[101].

These advances in electron waveshaping techniques raise the fundamental question of whether quantum electrodynamical (QED) interactions (e.g., light emission) can be controlled via electron waveshaping. To appreciate the importance of this question in practical applications, consider bremsstrahlung, the spontaneous emission of a free electron scattering off a static potential. Bremsstrahlung is responsible for the spectrally and angularly broad X-ray background from modern X-ray tubes. If QED interactions can indeed be controlled via electron waveshaping, bremsstrahlung could conceivably be made more directional, monochromatic, and versatile by structuring the emitting electron wavepacket, analogous to how radio waves are made more directional through structured emitters like phased-array antennas. This effect would be especially exciting in the hard X-ray regime, since the spatial resolution needed to manipulate the phases of hard X-rays cannot be achieved through material fabrication in optical elements, but is readily achievable through electron interference patterns.

In this paper, we present the concept of exploiting quantum interference in QED processes through shaped electron wavepackets, providing an additional degree of freedom in the design of these interactions. As an example, we apply our concept to bremsstrahlung. We show that it is possible to control spontaneous emission from a free electron through quantum interference enabled by electron waveshaping, just as spontaneous emission from an atom can be controlled through quantum interference between multiple atomic transitions[102] or through multiple atoms, as in superradiance[103]. Specifically, we show that free electron waveshaping can be used to tailor both the spatial

and the spectral distribution of the radiated photons. This results in enhanced directionality, monochromaticity, and versatility of photon emission compared to bremsstrahlung from an unshaped electron wavefunction. The ability to tailor the spatiotemporal attributes of photon emission via quantum interference provides additional degrees of freedom in shaping radiation across the entire electromagnetic spectrum. Beyond free electron radiation, the concept we present can be readily extended to processes involving more massive and non-elementary particles, such as neutrons, whose wavefunction can potentially be shaped as well[104–106].

## Results

In a general QED process, the transition probability from input state $|i_1\rangle$ to output state $|f\rangle$ is proportional to $|\langle f|S|i_1\rangle|^2 \equiv |\delta_1 M_1|^2$, where the S-operator $S$ transforms the quantum states at the start of the interaction into the quantum states at the end of the interaction. $\delta_1$ is the energy-conserving and/or momentum-conserving Dirac delta distribution, for scenarios with temporal and/or spatial translational invariance accordingly. $M_1$ is the scattering amplitude that abstracts away the part containing no delta distributions.

We begin by presenting a schematic approach that emphasizes the key points (full details are provided in the concrete examples we consider below). For an input state $|i_1 + i_2\rangle$, the cross section of the interaction is

$$\text{Cross section} \propto \int |\delta_1 M_1 + \delta_2 M_2|^2, \qquad (1)$$

where the integral in Eq. (1) is carried out over the output states, and $|i_1\rangle \neq |i_2\rangle$. For a general choice of input electron states, the Dirac delta distributions $\delta_1$ and $\delta_2$ peak at different combinations of output particle momenta, resulting in the cross terms canceling under the integration, i.e., the overall cross section can be written as an incoherent summation of cross sections (Fig. 1a):

$$\text{Incoherent cross section} \propto |M_1|^2 + |M_2|^2, \qquad (2)$$

where it is implicit that the various arguments in $M_1$ and $M_2$ have been assigned the values enforced through the integration of the respective Dirac delta distributions.

However, with precision particle waveshaping, it is possible to design a wavefunction composed of distinct input states such that $\delta_1$ and $\delta_2$ peak at the same combinations of output particle momenta, resulting in quantum interference between the scattering amplitudes associated with $|i_1\rangle$ and $|i_2\rangle$. This interference gives rise to coherent summation (Fig. 1b):

$$\text{Coherent cross section} \propto |M_1 + M_2|^2, \qquad (3)$$

which includes a nonzero—and generally substantial—contribution from the cross term $2\text{Re}\{M_1^* M_2\}$.

We see that the overall cross section is determined not only by the magnitudes of scattering amplitudes $M_1$ and $M_2$, but also by their relative phase, which can be controlled by the relative phase of states $|i_1\rangle$ and $|i_2\rangle$. The scenarios discussed in Eqs. (1)–(3) are readily extended to more than two input states, and reveal the ability of particle waveshaping to introduce an additional degree of freedom in the control of QED processes: namely, the design of QED processes not only through the magnitudes of the constituent scattering amplitudes, but also through the relative phases between these amplitudes. Our concept of harnessing quantum interference via electron waveshaping allows us to utilize the relative phases of scattering amplitudes for tailoring QED processes.

Comparing Eqs. (1)–(3) reveals the conditions to achieve spectrum control via wavefunction interference. This comparison also emphasizes why such possibilities have not been seen before.

**Fig. 1 Coherent and incoherent contributions in quantum electrodynamical (QED) processes.** When the input to a QED process is a superposition of multiple states, e.g., 2 states (Input A and Input B), the overall cross section is typically given by the sum of the cross sections associated with each input state, as in (**a**). Then, the total cross section is proportional to the sum of the squared-modulus of all the respective scattering amplitudes. However, when the input states are chosen to yield the same output state C, the individual processes coherently interfere, as in (**b**). The result is a square of summed amplitudes in (**b**), as opposed to the sum of squared amplitudes in (**a**). Essentially, **a** and **b** illustrate the concept that multiple quantum pathways will add coherently if and only if their output is the same, regardless of how much their input differ from one another. The specific case of bremsstrahlung is presented in (**c**) and (**d**) using Feynman diagrams, corresponding to the scenarios in **a** and **b**, respectively. The diagrams show the spontaneous emission of a photon from a free electron scattering off a static potential. The static potential is represented by the wiggly black line terminating in a cross. The coherent addition in **b** can be harnessed via free electron waveshaping as an additional degree of freedom to tailor the properties of QED processes. In bremsstrahlung, coherent interference **d** can lead to enhanced directionality, monochromaticity, and versatility in the photon output, as explored in Figs. 2 and 3.

For example, in Remez et al.[107] the emission of each photon was entangled to an outgoing electron, and the contributions to the emission from different initial electron angles could not interfere because each photon state was entangled to a different outgoing electron state. Similarly, work that considered shaping Cherenkov radiation through the OAM of electrons[90] found no change to the power spectrum, unless the outgoing electron was post-selected. We attribute these spontaneous emission results to the electron behaving ultimately as a point-like particle (as nicely put by Feynman[108]), regardless of its wavefunction. Even more recently, several other groups found that the emission intensity is not altered by the wavefunction in multiple scenarios[109–113]. This has led to the notion that only higher order correlation measurements (rather than the intensity) would be affected by the wavefunction of the emitter[114].

In stark contrast, we show here that the intensity can depend strongly on the emitter's wavefunction. This surprising result occurs when different contributions to an emitted photon state are entangled to the same outgoing electron state.

To exemplify the general concept in Fig. 1a, b we apply it to bremsstrahlung, the spontaneous emission of a photon by a free electron scattering off a static potential (Fig. 1c, d). We consider two examples for this potential: a neutral carbon atom in Fig. 2 and the magnetic field of a ferromagnet with nanoscale periodicity (i.e., a nano-undulator) in Fig. 3. The latter case is sometimes referred to as magnetic bremsstrahlung, or undulator radiation.

Now we consider an input electron wavepacket described as a superposition state $\int d^3\mathbf{p} \sum_s c_p^s |p, s\rangle$ composed of multiple states $|p, s\rangle$ (labeled by their four-momenta $p$ and spin $s$) weighted by complex coefficients $c_p^s$. We obtain the differential cross section (cross section $\sigma$ per unit angular frequency $\omega_{k'}$ per unit solid angle $\Omega_{k'}$) for an output photon of wavevector $k'$ as

$$\frac{d\sigma}{d\omega_{k'}d\Omega_{k'}} = \frac{c}{(2\pi)^4 vT} \int \frac{1}{\hbar^3} d^3\mathbf{p}' \frac{1}{4} \sum_{s',r'}$$

$$\left| \int d^3\mathbf{p} \left[ \delta\left(E_{p'} + \hbar\omega_{k'} - E_p\right) \sqrt{\frac{\hbar\omega_{k'}}{8\varepsilon_0 E_{p'}E_p}} \sum_s c_p^s M_{k'p'p}^{r's's} \right] \right|^2, \quad (4)$$

where the emission has been averaged over output spin $s'$ and

output photon polarization $r'$, and

$$M_{k'p'p}^{r's's} = -e^2 \bar{u}^{s'}(p') \tilde{A}_\nu \left(\frac{\mathbf{p}'}{\hbar} + \mathbf{k}' - \frac{\mathbf{p}}{\hbar}\right)$$
$$\left\{ \gamma^\mu \epsilon_{k',\mu}^{r'*} i \tilde{S}_F(p' + \hbar k')\gamma^\nu + \gamma^\nu i \tilde{S}_F(p - \hbar k')\gamma^\mu \epsilon_{k',\mu}^{r'*} \right\} u^s(p), \quad (5)$$

where $T$ is the interaction time, $e$ the elementary charge, $c$ the speed of light in free space, $\nu$ the speed of the input particles, $\varepsilon_0$ the permittivity of free space, $p^\mu$ ($\mu = 0, 1, 2, 3$) or simply $p$ the four-momentum for electrons (electron energy $E_p \equiv cp^0$), $k^\mu$ or simply $k$ the four-wavevector for photons (angular frequency $\omega_k \equiv ck^0$), and $\epsilon_k^{r,\mu}$ the polarization for a photon of wavevector $k$ and polarization $r$. Bold variables refer to the three-vector counterparts of the respective four-vectors, $\hbar$ is the reduced Planck's constant, $\gamma^\mu$ are the gamma matrices, and we use the repeated index convention $k^\mu p_\mu \equiv k^0 p^0 - \mathbf{k} \cdot \mathbf{p}$. Primes denote variables associated with outgoing particles. Based on the particles' dispersion relations, we have that $p^\mu p_\mu = m^2 c^2$, $k^\mu k_\mu = 0$, and $\epsilon_k^{r,\mu} k_\mu = 0$ ($m$ the electron mass). The u-type spinor is given by $u^s(p) = [\sqrt{p^\mu \sigma_\mu} \xi^s, \sqrt{p^\mu \bar{\sigma}_\mu} \xi^s]^T$, with column vectors $\xi^\uparrow = [1, 0]^T$ and $\xi^\downarrow = [0, 1]^T$ corresponding, respectively, to spin-up and spin-down. $\sigma^\mu = \{1, \sigma_x, \sigma_y, \sigma_z\}$ and $\bar{\sigma}^\mu = \{1, -\sigma_x, -\sigma_y, -\sigma_z\}$, $\sigma_{x,y,z}$ being the $2 \times 2$ Pauli matrices. Additionally, $\bar{u} = u^\dagger \gamma^0$ and $\tilde{S}_F(p) = \left(\gamma^\mu p_\mu - mcI\right)^{-1}$, $I$ being the $4 \times 4$ identity matrix. There are two kinds of photon polarizations, given by $\epsilon^{1,\mu} = \{0, 1, 0, 0\}$ and $\epsilon^{2,\mu} = \{0, 0, 1, 0\}$ in the case of a photon propagating in the $+z$ direction. In our calculations, we use the metric tensor and gamma matrix conventions of Peskin and Schroeder[115]. $\tilde{A}^\mu(\mathbf{k})$ refers to the Fourier transform of the static potential in the bremsstrahlung interaction. Details of the calculations leading to Eqs. (4) and (5) are given in the Methods section. Besides representing atomic potentials, $\tilde{A}^\mu(\mathbf{k})$ can be used to capture any type of static electromagnetic field, as well as time-dependent external fields by making $\tilde{A}^\mu$ a function of the full four-vector $k^\mu$.

From the energy-conserving delta distribution in Eq. (4), we see that quantum interference between the processes associated with different input states $p$ would occur if and only if the output of the various processes is identical. This further implies that the input states must have the same energy $E_p$ (and hence $|\mathbf{p}|$). We

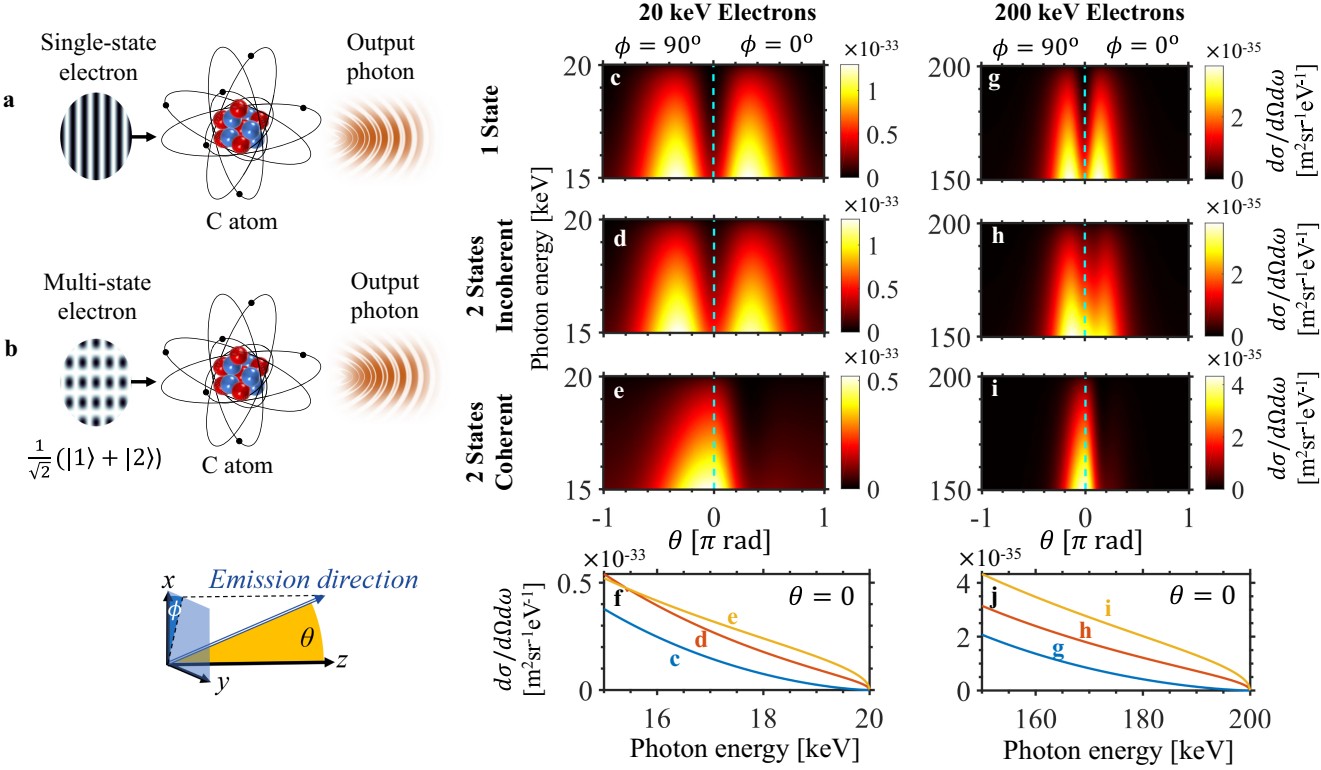

**Fig. 2 Enhanced directionality in atomic bremsstrahlung through shaped electron wavepackets.** In the typical atomic bremsstrahlung scenario **a**, a single momentum state electron scatters off a carbon atom and emits radiation. Shaping the input electron wavepacket through the use of multiple states as in **b**, where the input is a superposition of states |1⟩ and |2⟩, can enhance the output photon properties through coherent interference between the processes associated with each individual electron state. To illustrate this, **c–e** show the differential cross section $d\sigma/d\Omega d\omega$ of the emission process for 20 keV electrons, with a single $z$-directed input electron state in **c**, and two input electron states of opposite phase and oriented at $\theta_{1,2} = \pm15°$ with respect to the $z$-axis in **d** and **e**. A donut-shaped emission pattern, as indicated by the off-axis peaks, is expected for the single-state **c** and incoherent double-state **d** cases. In contrast, quantum interference between the constituent processes in **e** strongly suppresses off-axis emission, resulting in an emission pattern that is more directional and peaked on-axis. Cross-section emission patterns at $\theta = 0$ are compared in (**f**). The enhanced directionality also holds at other choices of electron energies and angles, as shown in **g–j**, which presents the emission spectra corresponding to the scenarios in **c–f**, respectively, but for 200 keV electrons, at $\theta_{1,2} = \pm15°$.

obtain the differential cross section in the case of coherent quantum interference as

$$\frac{d\sigma}{d\omega_{k'}d\Omega_{k'}} = \int d\Omega_{p'} \frac{1}{4} \sum_{s',r'} \left[ \frac{\omega_{k'}|\mathbf{p'}||\mathbf{p}|}{8\epsilon_0(\hbar c)^3(2\pi)^5} \left| \int d\Omega_p \sum_s c_p^s M_{k'p'p}^{r's's} \right|^2 \right],$$

(6)

where energy conservation $E_{p'} + \hbar\omega_{k'} - E_p = 0$ is implicitly enforced. Note that the differential rate $d\Gamma/d\omega_{k'}d\Omega_{k'}$ can be obtained from the differential cross section via the relation $d\Gamma/d\omega_{k'}d\Omega_{k'} = (v/V)(d\sigma/d\omega_{k'}d\Omega_{k'})$, $V$ being the interaction volume. We show quantitative results obtained using this formalism in Fig. 2.

Figure 2 considers bremsstrahlung radiation where the scattering potential $A^\mu$ is that of a neutral carbon atom, modeled using a sum of three Yukawa potentials fitted to the results of relativistic Hartree–Fock calculations, which agree well with experimental measurements[116,117] (see the Methods section). In all cases, the result is averaged over output spin and photon polarization, while the input electron states are taken to be spin-up. Figure 2a, b illustrates the two scenarios under consideration: an (unshaped) electron state of a single momentum traveling in the $+z$ direction, and a shaped electron input obtained by a superposition of 2 states, respectively. In the latter case, each of the two states have probability 0.5, a $\pi$ phase shift with respect to each other, and propagate at $\pm15°$ with respect to the $+z$

direction (i.e., shaped input $|i\rangle = (|p_+, \uparrow\rangle - |p_-, \uparrow\rangle)/\sqrt{2}$, where $\mathbf{p}_\pm \equiv p_0[\pm\sin\theta_i, 0, \cos\theta_i]^T$ and $\theta_i = 15°$). Such an input can be realistically generated using holography methods in electron microscopy, with a bi-prism or other analogs of double-slit experiments[53]. Note that the integral over the constituent momenta $\mathbf{p}$ of the incoming electron in Eqs. (4) and (6) is treated as a discrete sum over two states in this case. The electron kinetic energy of 20 keV is readily obtained from table-top scanning electron microscopes and from DC electron guns.

Figure 2c shows that the emission pattern for the single-state scenario is peaked off-axis (the plot range is cut at the highest possible output photon energy, equals the maximum kinetic energy of the input electron). Comparing Fig. 2d, e shows that there is a difference between the two possible emission patterns based on whether or not quantum coherent effects are considered. We see in Fig. 2e that the case of coherent superposition leads to more directional photon output compared to the single-state case in Fig. 2c, and also compared to the incoherent double-state case in Fig. 2d. Note that the latter is calculated by summing the cross-section results for the two momenta, as would be the case if this superposition state went through decoherence, reducing it to a mixed state of the two momenta of equal probability (i.e., the cross-terms resulting from the squared-modulus in Eq. (6) are ignored). To quantify the increased directionality in the quantum coherent case, we note that the ratio of the on-axis emission to the total emission in the shaped coherent case

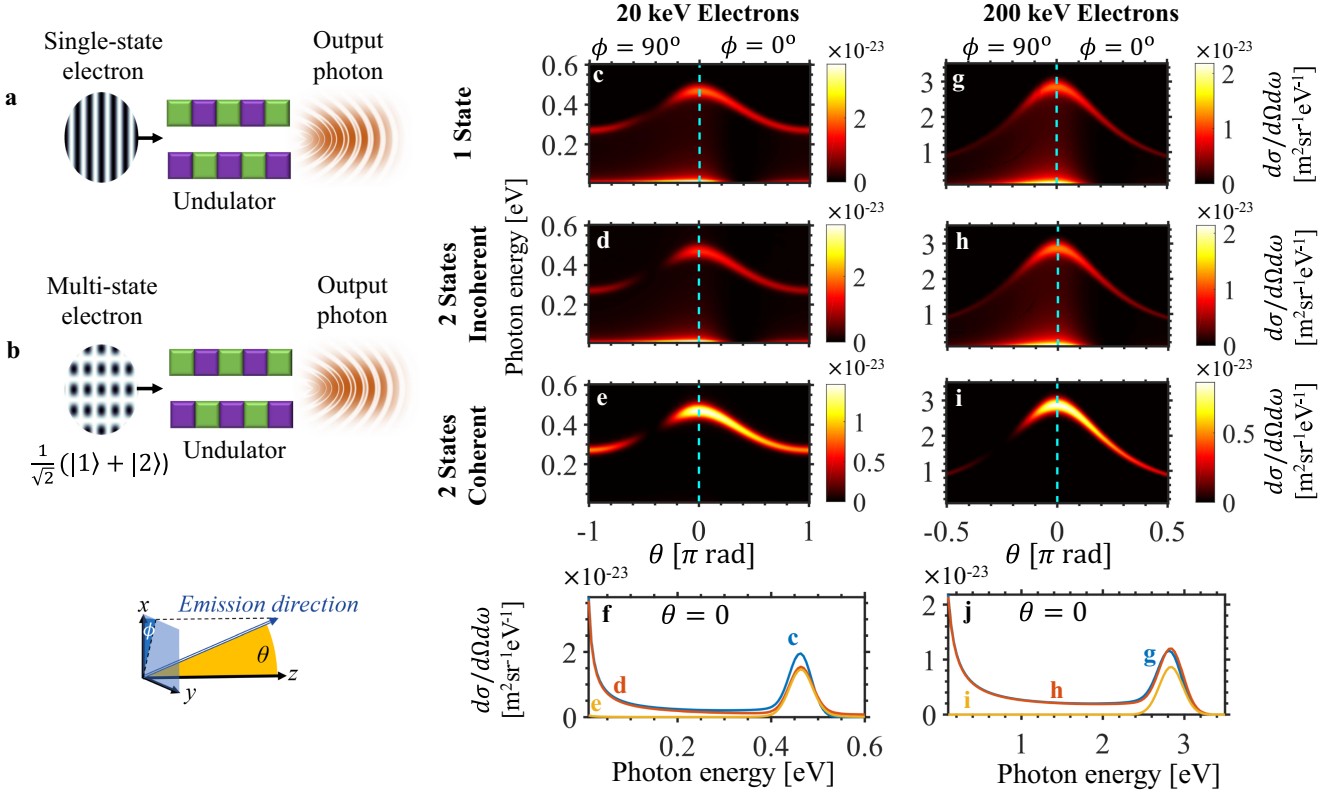

**Fig. 3 Enhanced monochromaticity of magnetic bremsstrahlung (undulator radiation) through shaped electron wavepackets.** We consider the scattering of an input electron off the magnetic field of a nano-undulator, for the case of a single momentum state input electron (**a**) and that of an input electron made up of two states |1⟩ and |2⟩ (**b**). **c** shows the differential cross section $d\sigma/d\Omega d\omega$ of the emission process for the single-state input electron scenario in (**a**). In addition to a relatively monochromatic peak, there is a strong synchrotron-radiation-like signature leading to relatively broadband radiation, with significant radiation components at lower photon energies. **d** and **e** show the emission patterns for the double-state input electron scenario in (**b**), with incoherent and coherent processes considered in (**d**) and (**e**), respectively. As **e** shows, the quantum coherence leads to destructive interference that strongly suppresses the broad synchrotron peak at low photon energies, leading to a more monochromatic output in a given direction. Cross sections of the emission patterns at $\theta = 0$ are compared in (**f**). The suppression of low photon energies continues to hold at other choices of electron energies, as shown in **g–j**, which presents the emission spectra for electrons of 200 keV, corresponding to the scenarios in **c–f**. The undulator considered is of period 1 μm and has an effective length of 5.3 μm. The two input electron states are of opposite phase and oriented at $\theta_{1,2} = \pm0.5°$ with respect to the z-axis in **d** and **e**, and at $\theta_{1,2} = \pm0.025°$ in **h** and **i**.

(Fig. 2e) is 3.27, a 12-fold enhancement of the corresponding ratio, 0.273, in the unshaped case (Fig. 2c). This quantum control of the bremsstrahlung output is directly related to the interference phenomenon described in Eqs. (1)–(3). To expound on this, we visually depict the coherent effects arising from electron waveshaping in Supplementary Information (SI) Section S1.

Figure 2g–i show that enhanced directionality can also be observed with 200 keV electrons, which are readily obtainable from table-top transmission electron microscopes and radiofrequency electron guns. In this case, the ratio of on-axis emission to total emission in the shaped coherent case (Fig. 2i) is 13.04, a 9-fold enhancement of the corresponding ratio, 1.404, in the unshaped case (Fig. 2g). We conclude that shaping the electron wavefunction enables significant control over the output angular distribution, as a direct result of quantum interference between different components of the wavefunction.

Beyond tailoring the spatial (angular) distribution of output radiation, quantum interference through electron waveshaping can also be harnessed to control the spectral (frequency) distribution of output photons. This is shown in Fig. 3, which explores bremsstrahlung from electrons scattering off the fields of a nano-undulator. Few-cycle undulators (also called wigglers) have potential applications in the generation of very short pulses of high-frequency light[118]. The study of a nano-undulator design

is also motivated by recent advances in nanofabrication of magnetic materials that can support large magnetic fields (~1 T) at the surface of nanopatterned ferromagnets[119–121], which has already drawn interest for applications like compact free electron X-ray sources[122]. We further explain and visually depict the role of quantum interference in enhancing nano-undulator radiation in Fig. S2 of SI Section S1.

The single-state and multi-state electron input scenarios are schematically illustrated in Fig. 3a, b. The output emission patterns for an electron with a kinetic energy of 20 keV are shown in Fig. 3c, f. Figure 3c corresponds to the single-state input scenario and shows that the main radiation peak in each direction is accompanied by a strong synchrotron-radiation-like signature, leading to relatively broadband radiation with significant radiation components at lower photon energies. Figure 3d, e correspond to radiation from two crossed electron states propagating at angles ±0.5° with respect to the +z direction, each having probability 0.5. In Fig. 3d the emission from the different momenta are summed incoherently, effectively describing a decohered superposition state.

If the two momentum states are coherent, however, interference in the emission channels must be taken into account, as we do in Fig. 3e. We observe that the radiation profile is significantly modified. In particular, the radiation is much more

monochromatic as the synchrotron-radiation-like tail in the low-photon-energy regime is greatly suppressed by destructive interference, whereas the undulator radiation peak remains relatively unaffected. Considering only photon energies above 0.01 eV, the total cross section in the shaped coherent case (Fig. 3e) is 16.5 $\mu m^2$, compared to the total cross section in the unshaped case (Fig. 3c) of 24.4 $\mu m^2$, indicating that electron waveshaping has reduced the total photon emission by 67.7%, with the majority of the suppression taking place at off-peak frequencies. To quantify this off-peak suppression, the coherent shaped electron's rate of emission is reduced 71-fold relative to the unshaped electron at photon energy 0.01 eV. The enhanced monochromaticity can be directly seen from the photon energy spread (standard deviation) decreasing by more than 10-fold from 62.5% in the unshaped case to 4.3% in the shaped case. In addition to the radiation being more monochromatic, the reduced photon emission at other unwanted frequencies reduces the rate of unwanted energy loss to radiation. Such a dependence on the electron wavepacket has intriguing consequences, as it can potentially lead to a longer mean free path for the electron in matter.

The enhanced monochromaticity induced by quantum interference can also be observed at other electron energies. For instance, Fig. 3g–j show the emission patterns for 200 keV input electron momentum states (propagating at angles ±0.025° with respect to the $+z$ direction), where we see that the suppression of the broadband synchrotron-radiation-like tail is even more pronounced. Considering only photon energies above 0.1 eV, the total cross section in the shaped coherent case (Fig. 3i) is 398.7 $\mu m^2$. We compare this value to the total cross section in the unshaped case (Fig. 3g) of 748.4 $\mu m^2$, indicating that electron waveshaping has reduced the total photon emission by 53.3%, with the majority of the suppression taking place at off-peak frequencies. Considering the emission at photon energy 0.1 eV, we observe a very large off-peak suppression with the shaped coherent photon's rate of emission (Fig. 3i) reduced 27,000-fold relative to the case with the unshaped electron (Fig. 3g). The enhanced monochromaticity can be directly seen from the photon energy spread (standard deviation) decreasing by more than 10-fold from 65.9% in the unshaped case to 3.6% in the shaped coherent case.

## Discussion

The predictions in our bremsstrahlung studies (Figs. 2 and 3) can be tested using microscopes with X-ray detectors, e.g., via energy-dispersive X-ray spectroscopy (EDS) or electron energy loss spectroscopy (EELS). The shaping of electron wavepackets can be accomplished through a variety of methods. The splitting of an electron into two interfering beams, for instance, can be achieved using an electron biprism[53,123] or single crystal thin films[124], allowing our examples to be studied using available technology. Other methods of electron beam shaping include the use of binary amplitude masks[89,95], patterned thin SiN membranes[97], electron–photon interactions[125], electrostatic elements[126,127], and magnetic vortices[128] and needles[129]. We detail proposals for experimentally demonstrating quantum control of bremsstrahlung processes via electron waveshaping in SI Section S2. Our calculations show that using table-top electron sources and realistic electron deflection angles from a biprism, we are able to obtain substantial and measurable photon emission rates, as well as significant measurable changes in emission characteristics moving from the unshaped to the shaped electron wavepacket cases.

We note that quantum interferences in QED that arise from electron wavepacket shaping can be understood as the free-electron QED analogs of coherent interference phenomena in atomic physics. Examples of such coherent phenomena are electromagnetically induced transparency[130], lasing without inversion[131], and refractive index enhancement[132], whereby intriguing physics arises from interference of the transition probability amplitudes between atomic states.

Furthermore, the enhancement of radiation using pre-shaped electron wavepackets explored here is highly complementary to other enhancement techniques, such as shaping the photonic density of states, e.g., with photonic crystals[133]. Some of the most famous enhancement techniques for free-electron radiation are related to the joint emission of multiple electrons, such as in self-amplified spontaneous emission (SASE)[134–138]. SASE involves the bunching of multiple charged particles through interaction with their own emitted radiation, and is an example of radiation enhancement via the shaping of the classical electron density distribution. Other examples include the shaping of electron density distributions via external means, such as nanoemitter arrays, magnets, and laser pulses[139–141]. In sharp contrast to these classical density shaping techniques, our presented mechanism leverages the wave nature of the electron wavepacket. As such, the radiation enhancement we predict can already occur at the level of a single charged particle, and does not require multiple particles. Shaping on a single-electron level is also qualitatively different from classical electron density shaping, as the latter has to manage inter-electron repulsion arising from the Coulomb force. In our work, it is noteworthy that just one electron constructed as a superposition of two momentum states can already lead to over 10 times more monochromatic radiation, as well as substantial reduction in unwanted radiation loss. It is also worth pointing out that the interference here is achieved by having different states along the direction perpendicular to the electron's propagation, and these effects would not be realized in a one-dimensional model of the electron.

Our findings provide a definitive answer to the fundamental question: can the quantum nature of the electron wavefunction affect the radiation it emits? When Schrödinger first introduced the quantum wavefunction, he interpreted it as the smooth charge density of a smeared-out particle[142]. Contradictions arising from this view eventually led to the interpretation of the wavefunction as a probability density of a point particle[143]; in the words of Feynman, "The electron is either here, or there, or somewhere else, but wherever it is, it is a point charge"[108]. Yet, intriguingly enough, it had been claimed that an electron behaves exactly like a smooth charge density in stimulated emission processes, which have been shown to depend on the waveshape of the emitting electron in both experiment[38] and semiclassical theory[144–146]. However, semiclassical theory does not capture spontaneous emission processes (which relies on the quantized nature of light), and so the fundamental question as to whether electron waveshaping can affect spontaneous emission had remained unanswered. The significance of a definitive answer to this question has been underscored by recent discussions in the context of shaping electrons for Cherenkov radiation (an example of a spontaneous emission process)[90], and a recent experiment that showed no dependence on wavefunction for Smith–Purcell radiation (yet another spontaneous emission process) in its regime of exploration[107]. Interestingly, recent findings also point out that when the electron is post-selected, spontaneous emission into near-field modes can depend on the symmetry of the initial electron wavefunction[129]. Through a fully quantum theory, we have now shown that fundamental principles support the notion that electron waveshaping can affect the emitted radiation. Furthermore, our theory shows that quantum interference (the coherent addition of multiple quantum pathways) is possible under special conditions, and can lead to drastic modifications in the radiation output.

In conclusion, we have presented the concept of engineering quantum interferences in QED processes through shaped electron wavepackets, providing an additional degree of freedom in the design and optimization of these processes. As an example, we applied our concept to bremsstrahlung, showing that it is possible to control this process of spontaneous emission from a free electron through the quantum interference resulting from electron waveshaping. Specifically, we show that free-electron waveshaping can be used to tailor both the spatial and the spectral distribution of the radiated photons, enhancing the directionality, monochromaticity, and versatility of photon emission compared to conventional bremsstrahlung. The reduced photon emission at unwanted frequencies and directions may help to reduce the rate of unwanted energy loss from the radiating electron, and thus potentially lead to a longer mean free path in matter for properly shaped electrons.

Looking forward, the concept presented in this work can be readily extended to QED processes with other charged particles like protons and ions, as well as processes in other field theories involving more elementary particles, such as pions, muons, and kaons, for which the same principles of waveshaping should apply. The prospect of coherent control over QED processes through particle waveshaping potentially opens up a wide vista of intriguing phenomena in fundamental and applied research, where the structure of electron wavepackets provides additional degrees of freedom to control and optimize electron-based quantum processes.

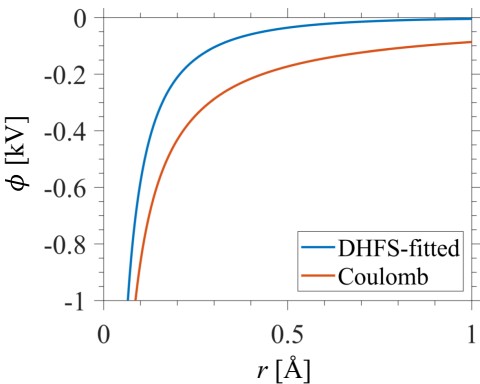

**Fig. 4 Electric potential of carbon atom.** We plot the DHFS-fitted scalar potential (blue) for the C atom used in our atomic bremsstrahlung calculations, with the Coulomb potential of the unshielded nucleus (red; Eq. (7) with the specified parameters, but setting $C_1 = 1$, $C_2 = \mu_1 = \mu_2 = 0$) for comparison.

## Methods

**Scalar potential of a neutral atom.** The carbon atom is modeled using a sum of three Yukawa potentials fitted to the results of the Dirac–Hartree–Fock–Slater (DHFS) self-consistent calculations as described in refs. [116,117]. In the space domain, this potential is given by

$$\phi_{\text{atom}}(\mathbf{r}) = \frac{-Ze}{4\pi\epsilon_0 |\mathbf{r}|} \sum_{j=1}^{2} C_j e^{-\mu_j \frac{|\mathbf{r}|}{a_0}}, \tag{7}$$

where $Z$ is the atomic number, $\epsilon_0$ the permittivity of free space, $a_0$ the Bohr radius and the constants $C_j$ and $\mu_j$ can be obtained from the tables in ref. [116]. For the neutral carbon atom, $Z = 6$, $C_1 = 0.1537$, $C_2 = 0.8463$, $\mu_1 = 8.0404$, and $\mu_2 = 1.4913$. For the neutral tungsten atom (tungsten being used in Fig. S4 of the Supplementary Information), $Z = 74$, $C_1 = 0.15$, $C_2 = 0.6871$, $C_3 = 1 - C_1 - C_2$, $\mu_1 = 28.63$, $\mu_2 = 4.2426$, and $\mu_3 = 1.234$. The Fourier transform of Eq. (7) yields

$$\tilde{\phi}_{\text{atom}}(\mathbf{k}) = \frac{-Ze}{\epsilon_0} \sum_{j=1}^{2} \frac{C_j}{|\mathbf{k}|^2 + \left(\frac{\mu_j}{a_0}\right)^2}. \tag{8}$$

The four-vector $\tilde{A}^\nu$ to be used in Eq. (5) is then $\tilde{A}^\nu(\mathbf{k}) = \{\tilde{\phi}_{\text{atom}}(\mathbf{k}), 0, 0, 0\}$. The scalar potential of a carbon atom in real space is shown in Fig. 4.

**Vector potential of a magnetic undulator.** The expression

$$\mathbf{\Pi}_{\text{und}}(\mathbf{r}) = \hat{\mathbf{x}} \frac{\Pi_0}{(2\pi)^2} \iint dk_z dk_y \, f(k_y, k_z) e^{ik_z z} e^{ik_y y} \frac{1}{2}(e^{q_x x} + e^{-q_x x}), \tag{9}$$

where $q_x \equiv \left(k_y^2 + k_z^2\right)^{\frac{1}{2}}$ and $\Pi_0$ is a constant prefactor, exactly solves the wave equation $\nabla^2 \mathbf{\Pi}_{\text{und}} = 0$, where $\nabla^2$ is the vector Laplacian. Treating $\mathbf{\Pi}_{\text{und}}$ as a Hertz potential (see, e.g., ref. [147]), we find that vector potential $\mathbf{A}_{\text{und}} = \mu_0 \nabla \times \mathbf{\Pi}_{\text{und}}$ and the magnetic flux density $\mathbf{B}_{\text{und}} = \mu_0 \nabla \times \nabla \times \mathbf{\Pi}_{\text{und}}$, $\mu_0$ being the permeability of free space. We can verify that $\nabla \times \mathbf{B}_{\text{und}} = \nabla \cdot \mathbf{B}_{\text{und}} = 0$, showing that Eq. (9) is a valid model for a general static magnetic field in free space. For instance, for the arbitrary scalar function $f = \delta(k_z - k_{z0})\delta(k_y - k_{y0})$, Eq. (9) gives the potential corresponding to a static magnetic field that is periodic over an infinite area in the $y$ and $z$ dimensions.

The Fourier transform of $\mathbf{A}_{\text{und}}$ is given by

$$\tilde{\mathbf{A}}_{\text{und}}(\mathbf{k}) = \mu_0 \Pi_0 f(k_y, k_z) \frac{1}{|\mathbf{k}|^2} \left[\hat{\mathbf{y}} ik_z - \hat{\mathbf{z}} ik_y\right] \times$$
$$\left[e^{\frac{q_x L}{2}}\left(q_x \cos\left(\frac{k_x L}{2}\right) + k_x \sin\left(\frac{k_x L}{2}\right)\right) + e^{-\frac{q_x L}{2}}\left(-q_x \cos\left(\frac{k_x L}{2}\right) + k_x \sin\left(\frac{k_x L}{2}\right)\right)\right], \tag{10}$$

where $q_x \equiv \left(k_z^2 + k_y^2\right)^{\frac{1}{2}}$ we terminate the undulator field at $x = \pm L/2$ since otherwise the undulator field blows up at large $x$, which is unphysical. In the $y$ and $z$ dimensions, we have used the profile

$$f(k_y, k_z) = \frac{1}{2}\left[\exp\left(-\frac{(k_z - k_{z0})^2}{\Delta_{k_z}^2} - \frac{(k_y - k_{y0})^2}{\Delta_{k_y}^2}\right) + \exp\left(-\frac{(k_z + k_{z0})^2}{\Delta_{k_z}^2} - \frac{(k_y + k_{y0})^2}{\Delta_{k_y}^2}\right)\right]. \tag{11}$$

In this study, we chose $k_{y0} = 0$, $k_{z0} = 6.28 \times 10^6$ m$^{-1}$ (corresponding to an undulator period of 1 μm in $z$), $\Delta_{k_y} = \Delta_{k_z} = 6.28 \times 10^5$ m$^{-1}$ (corresponding to 5.3 periods within the full-width-half-maximum of the on-axis magnetic field), and $L = 1$ μm. The magnetic field of this undulator is shown in Fig. 5.

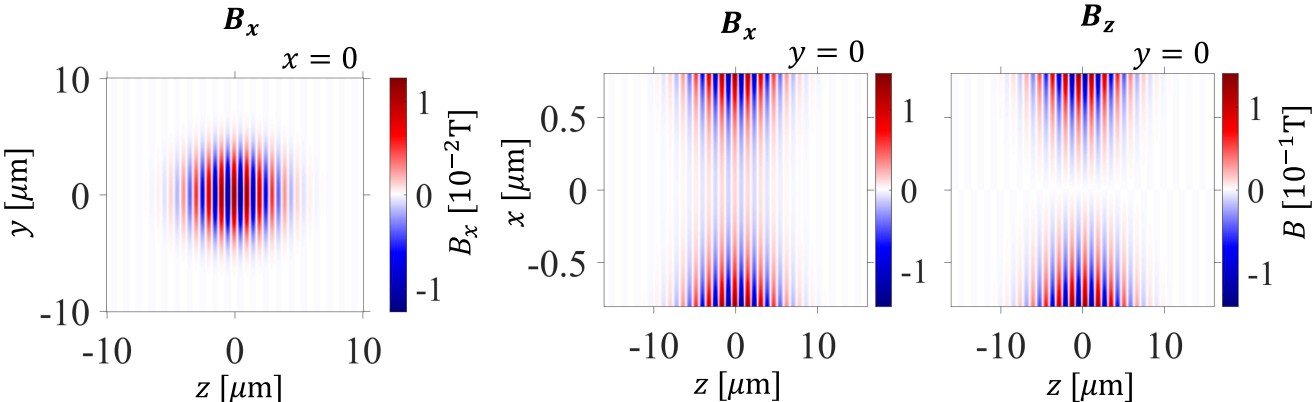

**Fig. 5 Magnetic field of nano-undulator.** To visualize the undulator used in our calculations, we plot the corresponding magnetic fields in the $x = 0$ and $y = 0$ planes.

**Differential cross section of bremsstrahlung with an electron wavepacket**. The time-evolution operator $U(t, t')$ evolves an initial state $|\Psi(t')\rangle$ to a final state $|\Psi(t)\rangle = U(t, t')|\Psi(t')\rangle$ (in the interaction picture) and is given by[115]

$$U(t, t') = \mathcal{T}\left\{\exp\left[-i\int_{t'}^{t}\frac{dt''}{\hbar}H_\mathrm{I}(t'')\right]\right\}, \; t \geq t', \quad (12)$$

where $\mathcal{T}$ is the time-ordering operator, and $H_\mathrm{I}$ is the interaction Hamiltonian for quantum electrodynamics (QED) in the interaction picture. Since bremsstrahlung is a second-order process, we can focus only on the second-order term in the expansion of Eq. (12), obtaining the scattering amplitude for shaped electron input $\int d^3\mathbf{p}\sum_s c_p^s|p, s\rangle$ as

$$S_{k'p'}^{r's'} = \int d^3\mathbf{p}\sum_s c_p^s\left\langle p', s'; k', r'\left|\left[-\frac{1}{2}\int_{t'}^{t}\frac{dt_1}{\hbar}\frac{dt_2}{\hbar}\mathcal{T}\{H_\mathrm{I}(t_1)H_\mathrm{I}(t_2)\}\right]\right|p, s\right\rangle, \quad (13)$$

where the output electron (of momentum $p'$ and spin $s'$) and photon (of wave-vector $k'$ and polarization $r'$) are represented by $|p', s'; k', r'\rangle$, and the input electron (of momentum $p$ and spin $s$) by $|p, s\rangle$. The electromagnetic field that the input electron scatters off is captured in $H_\mathrm{I}$ via the scattering potential $A^\mu$. We evaluate Eq. (13) to obtain

$$S_{k'p'}^{r's'} = \int d^3\mathbf{p}\, 2\pi\delta\left(E_{p'} + \hbar\omega_{k'} - E_p\right)\sqrt{\frac{\hbar c^4}{8V^3\epsilon_0\omega_{k'}E_{p'}E_p}}\sum_s c_p^s M_{k'p'p}^{r's's}, \quad (14)$$

where we have taken the limit of infinite interaction time $T \to \infty$. The cross section, averaging over output electron spin and output photon polarization, is then

$$\sigma = \frac{V^3}{vT}\int\frac{d^3\mathbf{p}'}{(2\pi\hbar)^3}\frac{d^3\mathbf{k}'}{(2\pi)^3}\frac{1}{4}\sum_{s', r'}\left|S_{k'p'}^{r's'}\right|^2, \quad (15)$$

from which we readily obtain Eq. (4) by considering the differential cross section with respect to the output photon momentum.

## Data availability

All data that support the plots and other findings within this paper are available from the corresponding authors on reasonable request. Source data are provided with this paper.

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

## Acknowledgements

The authors acknowledge helpful discussions with Prof. Max Tegmark, Prof. Robert L. Jaffe, and Prof. Nahid Talebi. This material is based upon work supported in part by the U.S. Army Research Office through the Institute for Soldier Nanotechnologies at MIT, under Collaborative Agreement Number W911NF-18-2-0048; by the Air Force Office of Scientific Research under the award number FA9550-20-1-0115; as well as by the Defense Advanced Research Projects Agency (DARPA) under Agreement No. HR00112090081. This research was supported by the Agency for Science, Technology and Research (A*STAR) Science & Engineering Research Council (Grant No. A1984c0043), and the Binational USA–Israel Science Foundation (BSF) grant number 2018288. L.J.W. acknowledges the support of the Nanyang Assistant Professorship Start-up Grant. N.R. was supported by Department of Energy Fellowship DE-FG02-97ER25308, and a Dean's Fellowship by the MIT School of Science.

## Author contributions

L.J.W., N.R., C.M., T.C., J.D.J., M.S., and I.K. were involved in the conception and design of the work. L.J.W., N.R., C.M., and I.K. performed the calculations and analysis. All authors were involved in interpreting the results and drafting the work.

## Competing interests

The authors declare no competing interests.
