## [Peer Review File · Nature Communications]

Reviewers' Comments:

Reviewer #1 (Remarks to the Author)

In this manuscript the authors develop a theoretical understanding of how electron wavefunction shaping can change the spatial properties of generated radiation in a broad region of the electromagnetic spectrum. In particular, the authors optimize the collimation of radiation from electron beams passing through spatially periodic electromagnetic fields of an undulator. The studies suggest that the radiation from rapidly decelerated shaped electrons can be directional in space, as well as spectrally monochromatic. The theoretical investigations extend further to show that a tunable radiation source may be feasible over wide range of frequencies including the optical range and potentially spanning all the way to the X-ray regime. The concept is of great interest to the synchrotron community and may greatly reduce flux losses. This concept may be applicable to other non-coherent or partially coherent sources as well. It is hard to judge the validity of the proposed approach without any experimental evidence. I would be glad to review the manuscript positively if supported by any demonstration in a region of the spectrum which is relatively easily accessible.

Reviewer #2 (Remarks to the Author)

The paper deals with radiation patterns from electrons whose wavefunction is in some non-trivial configuration on the scale of the radiation wavelength.

While the results are interesting I am not convinced they are valid. As far as I can tell, the approach is to take the charge density associated with the wavefunction as a sort of a rigid body. In reality a quantum mechanical wavefunction, unless it is a stationary state, will spread, even if there is no applied field at all. If the authors indeed neglected wavefunction spreading, what is the justification for this?

On a related matter, it is not true, generally, that one can Lorentz transform away the relativistic nature of a given wavefunction. The wavefunction may have a broad spectrum of momenta. If the momentum spread is relativistic, then there is no frame where all the relativistic components disappear.

For the above reasons, before considering the manuscript further, I would ask for clarification on how the wavefunction is evolved in these calculations, and how the procedure is justified.

Reviewer #3 (Remarks to the Author)

This manuscript proposes a potential technique to control (tailor) the electromagnetic field radiation with a wave function (wave-packet) shaping of an emitter electron in the quantum mechanical phase space, which leads to tailoring an electron space charge density distribution in the macroscopic phase space. As application examples of this technique, authors address undulator radiation with wavelength of 1 nm (1 keV photon energy) reminiscent of x-ray SASE FEL [R. Bonifacio and F. Casgrande, Nucl. Instr. Methods A237, 168 (1985); R. Bonifacio et al., Phys. Rev. Lett. 73, 70 (1994); A. Serbeto et al., Phy. Plasmas 15, 013110 (2008); C. B. Schroeder et al., Phys. Rev. E64, 056602 (2001); E.L. Saldin et al., Nucl. Instr. Methods A475, 357 (2001)], optical radiation with wavelength of 600 nm using an electron microscope electron source and the production of directional, monochromatic Bremsstrahlung using a capacitor structure such as a photonic bandgap (PBG) structure [A. Chutinun et al., Phys. Rev. Lett. 90, 123901 (2003)], although the novelty and superiority of each application have not been mentioned in comparison with the well-known theory and technology particularly for x-ray FEL as well as unclear physical basis on a doubtful wave function obtained from physically unrealistic too-much-simplified Schrodinger equation and radiation formalism. The most evident verification of this proposal

should be demonstrated by the experimental results.
In this context, this manuscript would not be recommended for the publication.

Reviewer #4 (Remarks to the Author)

The paper presents work that seems interesting, however, it is at a rather preliminary stage. The main finding of the paper is included in Section III: setting a charge density in (2), the authors determine the Poynting vector and observe its enhanced directivity; subsequently, they consider two applications in IV and V.

It is not clear how these simplified case studies would play out when the charge distribution is realized; to that end, simulations of actual devices with appropriate boundary conditions, i.e. a more realistic mathematical framework than just solving the Schroedinger equation, would be necessary.

Without either realistic simulations, or experimental results, the observation that the solutions to Schroedinger's equations can be made directive for a sinusoidal charge distribution does not merit publication in Nature Comms (in the reviewer's opinion). In fact, this observation is almost evident, considering the volume of work on synthesizing antenna current distributions with directive/super-directive far fields and the analogy that one can draw from the one case to the other.

Finally, one interesting direction would be to synthesize ρ_v for a specific pattern of the Poynting vector, i.e. the inverse problem.

Response to Referees

Reviewer #1 (Remarks to the Author):

In this manuscript the authors develop a theoretical understanding of how electron wavefunction shaping can change the spatial properties of generated radiation in a broad region of the electromagnetic spectrum. In particular, the authors optimize the collimation of radiation from electron beams passing through spatially periodic electromagnetic fields of an undulator. The studies suggest that the radiation from rapidly decelerated shaped electrons can be directional in space, as well as spectrally monochromatic. The theoretical investigations extend further to show that a tunable radiation source may be feasible over wide range of frequencies including the optical range and potentially spanning all the way to the X-ray regime. The concept is of great interest to the synchrotron community and may greatly reduce flux losses. This concept may be applicable to other non-coherent or partially coherent sources as well. It is hard to judge the validity of the proposed approach without any experimental evidence. I would be glad to review the manuscript positively if supported by any demonstration in a region of the spectrum which is relatively easily accessible.

Reply: We thank the referee for acknowledging the interest of communities such as the synchrotron community in our work. We emphasize that to date, there is no example in the literature of radiation enhancement through quantum interference caused by electron waveshaping. Such a treatment has remained missing to this day in spite of rapidly growing interest in electron waveshaping techniques. Due to the novelty of the idea and its potential importance, we believe that a rigorous theoretical treatment is a valuable contribution.

When we submitted the original manuscript several years ago, it only had a simplified theory (semiclassical) treatment, which may have indeed seemed too simplified to be trusted without a supporting experiment. Consequently, we have revised the manuscript extensively to change the underlying method to a *fully* quantum electrodynamical (QED) description, which amounts to an *ab initio* description of the system.

The basic exciting idea remained exactly the same, but is now supported by an exact theoretical treatment, which we believe makes the manuscript much stronger, and valuable to help lead to an experimental observation. The formalism we use has been responsible for many famous breakthroughs in physics, from the Lamb shift to Compton scattering, yet this is the first time anyone has studied this aspect of the formalism, which yields sweeping revelations on the role of particle waveshaping in quantum processes.

Specifically, we present a framework based on second quantization (Feynman diagrams) that employs a relativistic Dirac description for the input fermion states. We then calculate the resulting photon emission from time-dependent perturbation theory, taking into account the coherent interference of S-matrix elements, which lie at the heart of the predicted phenomena. Our findings verify our original conclusions qualitatively, but also results in interesting quantitative changes. Most importantly, the new formalism confirms that electron wavefunction shaping can indeed control the spatial and spectral properties of generated radiation in a broad region of the electromagnetic spectrum.

Based on the referee's suggestions, we have also revised our manuscript to directly showcase an example of enhanced monochromaticity in an undulator, where we see for example that waveshaping can indeed reduce the unwanted radiation substantially. This leads to more monochromatic output in each a given direction and also greater overall efficiency if the electrons are recycled. In our examples, we also use electron energies accessible from scanning electron microscopes (SEMs) and transmission electron microscopes (TEMs), showing that this enhanced monochromaticity via waveshaping can be readily demonstrated using lab-scale electron sources. In light of these revisions, we hope that the referee will be able to give a favorable recommendation for the acceptance of our work.

Reviewer #2:

The paper deals with radiation patterns from electrons whose wavefunction is in some non-trivial configuration on the scale of the radiation wavelength.

While the results are interesting I am not convinced they are valid. As far as I can tell, the approach is to take the charge density associated with the wavefunction as a sort of a rigid body. In reality a quantum mechanical wavefunction, unless it is a stationary state, will spread, even if there is no applied field at all. If the authors indeed neglected wavefunction spreading, what is the justification for this?

Reply: We thank the referee for raising the above points. To account for all aspects, we have revised the manuscript to change the underlying method to a *fully* quantum electrodynamical (QED) description, which amounts to an *ab initio* description of the system.

Specifically, we present a framework based on second quantization (Feynman diagrams) that employs a relativistic Dirac description for the input fermion states. We then calculate the resulting photon emission from time-dependent perturbation theory, taking into account the coherent interference of S-matrix elements, which lie at the heart of the predicted phenomena.

Our QED theory now rigorously accounts for the spreading of a wavepacket by requiring that the input electron wavepacket be built from multiple input momentum states. In our examples, we consider two input momentum states, which can be realistically generated, for example using holography methods in electron microscopy, with a bi-prism or other analogues of double-slit experiments [Tonomura et al., Am. J. Phys 57, 117-120 (1989)].

Our findings verify all our original qualitative predictions but with quantitative corrections in several places. These changes show that the referee was right to question our assumption (e.g., treating the wavefunction as a rigid body creating a charge distribution), and we are grateful to the referee for motivating us to improve our work in this direction. What is most important for us is that the new rigorous treatment now confirms that electron wavefunction shaping can indeed control the spatial and spectral properties of generated radiation in a broad region of the electromagnetic spectrum.

On a related matter, it is not true, generally, that one can Lorentz transform away the relativistic nature of a given wavefunction. The wavefunction may have a broad spectrum of momenta. If the momentum spread is relativistic, then there is no frame where all the relativistic components disappear.

Reply: We thank the referee for raising this point and we completely agree that this was missing in our original treatment of the problem. The framework we now present is based on fully relativistic QED and therefore makes no assumption regarding the ability to Lorentz transform away the relativistic nature of wavefunctions.

For the above reasons, before considering the manuscript further, I would ask for clarification on how the wavefunction is evolved in these calculations, and how the procedure is justified.

Reply: We have replaced our semiclassical theory with a fully QED, *ab initio* approach. We have resubmitted the manuscript since the rigorous approach confirms the strong predictions of our original study. In light of these revisions, we hope that the referee will share our excitement about these predictions and support our work further.

Reviewer #3:

This manuscript proposes a potential technique to control (tailor) the electromagnetic field radiation with a wave function (wave-packet) shaping of an emitter electron in the quantum mechanical phase space, which leads to tailoring an electron space charge density distribution in the macroscopic phase space. As application examples of this technique, authors address undulator radiation with wavelength of 1 nm (1 keV photon energy) reminiscent of x-ray SASE FEL [R. Bonifacio and F. Casgrande, Nucl. Instr. Methods A237, 168 (1985); R. Bonifacio et al., Phys. Rev. Lett. 73, 70 (1994); A. Serbeto et al., Phy. Plasmas 15, 013110 (2008); C. B. Schroeder et al., Phys. Rev. E64, 056602 (2001); E.L. Saldin et al., Nucl. Instr. Methods A475, 357 (2001)], optical radiation with wavelength of 600 nm using an electron microscope electron source and the production of directional, monochromatic Bremsstrahlung using a capacitor structure such as a photonic bandgap (PBG) structure [A. Chutinun et al., Phys. Rev. Lett. 90, 123901 (2003)], although the novelty and superiority of each application have not been mentioned in comparison with the well-known theory and technology particularly for x-ray FEL as well as unclear physical basis on a doubtful wave function obtained from physically unrealistic too-much-simplified Schrodinger equation and radiation formalism. The most evident verification of this proposal should be demonstrated by the experimental results. In this context, this manuscript would not be recommended for the publication.

Reply: We thank the referee for raising these points and suggesting these references that we now cite. We have now thoroughly revised the manuscript – presenting a strong *ab initio* theory. Using this theory, we clarify the core difference between our proposal and classical bunching of point charged particle, as is the case for SASE in X-ray FELs. The difference is now explained in the revised manuscript, where we also bring specific examples in a context relevant to undulator radiation.

Let us explain the core difference between our work and the effect of bunching: our presented mechanism leverages the wave nature of the electron wavepacket, rather than the classical charge distribution. We also show that the influence on the emitted radiation is critically different. For one, the radiation enhancement we predict can already occur at the level of a single charged particle, and does not require multiple particles. For example, we show that even just one electron constructed as a superposition of two momentum states can already lead

to over 10 times more narrowband radiation as well as a substantial reduction in unwanted radiation loss.

We emphasize that to date, there is no example in the literature of radiation enhancement through quantum interference caused by electron waveshaping. Such a treatment has remained missing in spite of rapidly growing interest in electron waveshaping techniques. Due to the novelty of the idea, we believe that it is a valuable contribution to many communities, including electron microscopy, free electron radiation, and X-ray science.

The important prospects and the surprising results motivated us to now develop a *fully* quantum electrodynamical (QED) formalism for radiation from a shaped electron wavepacket. We believe that such an *ab initio* description of the system can help design and motivate the first experiments that can test the idea.

Specifically, we present a framework based on second quantization (Feynman diagrams) that employs a relativistic Dirac description for the input fermion states. We then calculate the resulting photon emission from time-dependent perturbation theory, taking into account the coherent interference of S-matrix elements, which lie at the heart of the predicted phenomena. Our findings verify our original conclusions qualitatively, but also results in interesting quantitative changes. Most importantly, **the new formalism confirms that electron wavefunction shaping can indeed control the spatial and spectral properties of generated radiation** in a broad region of the electromagnetic spectrum.

Placing our new contribution in a historical context, the second quantization formalism of QED has been responsible for many famous breakthroughs in physics through the years, from the Lamb shift to Compton scattering, yet this is the first time anyone has applied this formalism to study the role of particle waveshaping in quantum processes, which has yielded sweeping revelations on the use of shaped wavepackets to design and enhance light-matter interaction such as free electron radiation.

More information regarding the references suggested by the referee:

Our structuring of the electron wavepacket is highly complementary to external structure-based enhancement techniques such as the PBG paper by Chutinan et al, as well as self-induced enhancement such as SASE, and we have made sure to clarify this in the text, where we also cited all of the references this referee has provided.

Thanks to the helpful feedback of this and the other referees, we are now able to present the first full quantum electrodynamical (QED) theory on the control of quantum processes via free electron waveshaping. In light of the fact that such a treatment has remained missing from the literature to this day, we hope that the referee will share our excitement about our new formalism and its strong consequences.

Reviewer #4:

The paper presents work that seems interesting, however, it is at a rather preliminary stage. The main finding of the paper is included in Section III: setting a charge density in (2), the authors determine the Poynting vector and observe its enhanced directivity; subsequently, they consider two applications in IV and V.

It is not clear how these simplified case studies would play out when the charge distribution is realized; to that end, simulations of actual devices with appropriate boundary conditions, i.e. a more realistic mathematical framework than just solving the Schroedinger equation, would be necessary.

Without either realistic simulations, or experimental results, the observation that the solutions to Schroedinger's equations can be made directive for a sinusoidal charge distribution does not merit publication in Nature Comms (in the reviewer's opinion). In fact, this observation is almost evident, considering the volume of work on synthesizing antenna current distributions with directive/super-directive far fields and the analogy that one can draw from the one case to the other.

Reply: We thank the referee for raising these points, and for recommending a more realistic mathematical framework. In accordance with his/her comments, we have now replaced our semiclassical theory with a rigorous, relativistic quantum electrodynamical (QED) approach that confirms the essential conclusions of our original study. Such a treatment has remained missing from the literature to this day in spite of rapidly growing interest in electron waveshaping techniques.

With regards to the observation that the idea is analogous to synthesizing antenna current distributions, we note first of all that this powerful analogy has already led to many major topics (no less important due to this analogy) in physics, including superradiance in quantum optics and the self-bunching of electrons in free electron lasers via self-amplified spontaneous emission (SASE). Furthermore, the fully quantum treatment in our revised manuscript shows that this coherent enhancement of QED processes can now **go far beyond the direct analogy to conventional antenna theory**, as it involves the wave nature of electrons. Whereas antennas require more than 1 physical element to achieve emission enhancements, we show that even just one electron constructed as a superposition of two momentum states can already lead to over 10 times more narrowband radiation as well as a substantial reduction in unwanted radiation loss.

Finally, one interesting direction would be to synthesize $\rho_{\mathbf{v}}$ for a specific pattern of the Poynting vector, i.e. the inverse problem.

Reply: We thank the referee for this suggestion. Indeed, the inverse problem is exciting to consider, and can have significant importance for optimizing the shaping that will be necessary for finding the best quantitative performance of X-ray generation from shaped electron wavepackets. We are keen on this as a future direction, and believe that the fully QED framework we have established in our work builds the groundwork for such a pursuit.

Following the thorough revisions that we have made in our manuscript, we hope that the referee can consider our work in a favorable light.

Reviewers' Comments:

Reviewer #5:

Remarks to the Author:

In this work, the authors theoretically study how to control the quantum electrodynamical processes by shaping electron wavepackets. As an example, the concept is applied to Bremsstrahlung revealing the enhanced directionality and monochromaticity of photon emission. The work is interesting and the results should attract the attentions in the field of free-electron laser, electron microscopes, materials analysis and so on. By using an ab initio description of the system, it seems that the rigorous theoretical results are credible. However, some questions should be answered before making the final decision.

(1) The enhanced directionality and monochromaticity of photon emission are shown for Bremsstrahlung by free electron wave-shaping. Since the electron density distribution can also control the spatial and the spectral distribution of photon emission, what's the different between the wave-shaping and electron density distribution control if the free electron density does not decreased to a rather low value? Can enhanced directionality and monochromaticity be realized by controlling the electron density distribution?

(2) The calculation of two examples (atomic Bremsstrahlung and magnetic Bremsstrahlung) is convincing and precisely confirms the enhancement of directionality and monochromaticity. But the fundamental mechanism of this improvement is not well explained and the physical process is not very unequivocal. More detailed analysis of the entire physical process is needed.

Reviewer #6:

Remarks to the Author:

In the manuscript by Wang et al., the authors introduce a rigorous theoretical formalism aimed at demonstrating that the rate of quantum electrodynamical (QED) interactions, which lead to photon emission, can be controlled via electron wave shaping. In particular, they show that free-electron wave shaping can be used to tailor both the spatial and the spectral distribution of the emitted photons. To demonstrate this, the authors applies the results of their theory to the illustrative the case of Bremsstrahlung. Spatial tailoring is theoretically predicted by analyzing the interaction between an electron in a superposition of multiple (i.e. two) momenta and a Carbon atom. Such an interaction leads to an enhanced directionality of the emitted photons. Similarly, spectral tailoring is theoretically verified by analyzing the interaction between an electron (in a superposition of multiple, i.e. two, momenta) and an undulator system. Such a process leads to an enhanced monochromaticity and directionality of the emitted photons.

The authors claim to have demonstrated for the very first time that the properties of the photons emitted in QED processes can be tailored through shaping of the electron wave packet. This is somehow unprecedented and of high interest and impact for several applications, such as energy dispersive X-ray spectroscopy (EDS) and electron energy loss spectroscopy (EELS). The applicability of the proposed theory is indeed testified by the fact that the authors consider the potential practical example of Bremsstrahlung, which is routinely employed in state-of-the-art X-ray sources for medical imaging, security scanning, materials analysis, and astrophysics. The new approach introduced in this manuscript can thus open novel avenues to technologies that are based on free-electron wave-shaping (for example, medical imaging, industrial quality inspection, and security scanning).

As can be inferred from the referees' comments, as well as from the authors' response, the previous version of the manuscript was significantly different from the current one. In particular, in a first stage, the authors made use of a semiclassical approach based on solving the Schrödinger equation for the electron wavefunction in order to assert their claim. However, some reviewers (especially, referee #2) questioned such a semiclassical treatment, since it did not (and could not) provide a rigorous and solid theoretical demonstration of all their findings. The revised manuscript now presents a complete and more meticulous analysis that is entirely based on the QED theory and that provides strong theoretical evidence of their claims. I believe that the new version of the manuscript definitely

raises the level of the entire work.

Yet, despite its excellence and potential high impact, this manuscript still presents what we believe is a relevant issue, i.e. the lack of an experimental observation supporting the new developed theory, as already pointed out by all the other reviewers (especially by referee #1). A valid experiment accompanying and evidencing the theory is, we believe, typically expected in journal such as Nature Communications. The authors mention throughout the text some practical methods currently available to generate the required electron beams. In particular, they stress the fact that their theory could be directly applied in a simple fashion to electron energies provided by sources commonly feeding scanning electron microscopes (SEMs) and transmission electron microscopes (TEMs). However, such asserts cannot be in principle considered, in our opinion, either as an exhaustive argument or a convincing experimental evidence. Furthermore, the authors do not explain or suggest how to achieve wave shaping in a practical scenario, even though they mention some experiments in which their outcomes could be experimentally verified. Overall, the lack of an experimental evidence makes the manuscript difficult to assess.

In conclusion, the manuscript introduces a novel methodology that could be of high interest, potentially bringing about a great impact for scientists working in the specific fields of electron wave-shaping applications, as well as for related technologies (medical imaging, industrial quality inspection, and security scanning). However, as an experimental investigation supporting the theoretical analysis is missing, the manuscript would probably be of higher value and impact with some experimental evidence or perhaps more suitable for a journal like Phys Rev Letter. Nevertheless, I am not contrary to the publication in Nature Communications, if the Editor is ready to take some risk in terms of future feasibility, as the theoretical approach is very interesting per se.

Response to Referees

We thank the referees for their positive responses and helpful comments. We provide a point-by-point reply to their remarks below

Reviewer #5 (Remarks to the Author):

In this work, the authors theoretically study how to control the quantum electrodynamical processes by shaping electron wavepackets. As an example, the concept is applied to Bremsstrahlung revealing the enhanced the directionality and monochromaticity of photon emission. The work is interesting and the results should attract the attentions in the field of free-electron laser, electron microscopes, materials analysis and so on. By using an ab initio description of the system, it seems that the rigorous theoretical results are credible. However, some questions should be answered before making the final decision.

Authors' reply: We thank the referee for the positive feedback and remarks.

(1) The enhanced directionality and monochromaticity of photon emission are shown for Bremsstrahlung by free electron wave-shaping. Since the electron density distribution can also control the spatial and the spectral distribution of photon emission, what's the different between the wave-shaping and electron density distribution control if the free electron density does not decreased to a rather low value? Can enhanced directionality and monochromaticity be realized by controlling the electron density distribution?

Authors' reply: We thank the referee for raising this point, which we now also address in the revised manuscript. We note the mechanism we present in our paper is highly complementary to radiation enhancement techniques that involve the shaping of the classical electron density distribution. These latter techniques include self-amplified spontaneous emission (SASE) from micro-bunched electron pulses and electron density shaping using nanoemitter arrays, magnets and laser pulses. These techniques can indeed also achieve enhanced directionality and monochromaticity in the resulting radiation, as happens for example in free-electron lasers.

However, there is a fundamental difference between these classical density shaping techniques and our proposal – our presented mechanism leverages the wave nature of the electron wavepacket. As such, the radiation enhancement we predict can already occur at the level of a single charged particle, and does not require multiple particles.

Shaping on a single-electron level is qualitatively different from classical electron density shaping. We show in the revised manuscript that calculating the radiation from the electron wavefunction as if it describes a classical electron density produces different predictions (and thus of course only the full *ab initio* description predicts the correct physics).

Furthermore, the case of a multi-electron distribution has to consider inter-electron repulsion arising from the Coulomb force. In contrast, it is noteworthy that just one electron constructed as a superposition of two momentum states can already lead to over 10 times more monochromatic radiation as well as a substantial reduction in unwanted radiation loss.

This difference could have critical practical applications, as it is currently not possible to shape electron densities on scales relevant for X-ray radiation, except inside enormous facilities such as free-electron lasers, where electrons are ultra-relativistic. Our work enables one to achieve similar benefits in much simpler experimental setups such as electron microscopes.

We have added a passage containing these details in the discussion (specifically, the 4th-to-last paragraph) to ensure they are clear to the reader.

(2) The calculation of two examples (atomic Bremsstrahlung and magnetic Bremsstrahlung) is convincing and precisely confirms the enhancement of directionality and monochromaticity. But the fundamental mechanism of this improvement is not well explained and the physical process is not very unequivocal. More detailed analysis of the entire physical process is needed.

Authors' reply: We thank the referee for making this comment. To ensure that the fundamental mechanism of the improvement is well explained, and to provide detailed analysis, we have added a new Supplementary Information (SI) Section (Section S1), as well as two illustrative figures (Figs. S1 and S2) explaining in detail how quantum interference arising from electron waveshaping leads to the observed enhancements. We have also modified and extended the relevant segments of the main text to provide a clear explanation.

Figure S1. Illustration of quantum interference arising from a two-state electron input, resulting in enhanced atomic bremsstrahlung. The parameters of this study are exactly those used in Fig. 2g-j of the main text. (a) shows Fig. 2i rendered in a different colormap, where quantum interference between the constituent processes results in an output cross section $|A + B|^2$ (A and B correspond to the transition amplitudes from the two input states respectively). To understand the contrast between the coherent output $|A + B|^2$ vs. the incoherent output $|A|^2 + |B|^2$, we break $|A + B|^2 = |A|^2 + |B|^2 + 2\text{Re}\{A^*B\}$ down into its constituent terms in (b-d). We see that the cross term shown in (d) is instrumental in the suppression of off-axis radiation, as well as in the enhancement of on-axis radiation, in the final result (a). This represents destructive interference off-axis, but constructive interference on-axis between the processes A and B .

Figure S2. Illustration of quantum interference arising from a two-state electron input, resulting in enhanced undulator bremsstrahlung. The parameters of this study are exactly those used in Fig. 3c-f. (a) shows Fig. 3e rendered in a different colormap, where quantum interference between the constituent processes results in an output cross section $|A + B|^2$ (A and B correspond to the transition amplitudes from the two input states respectively). To understand the contrast between the coherent output $|A + B|^2$ vs. the incoherent output $|A|^2 + |B|^2$, we break $|A + B|^2 = |A|^2 + |B|^2 + 2\text{Re}\{A^*B\}$ down into its constituent terms in (b-d). We see that the cross term shown in (d) is instrumental in the suppression of low-frequency radiation in the final result (a). This represents destructive interference at lower frequencies, but not at higher frequencies near the desired peak, between the processes A and B .

Once again, we thank the referee for the highly encouraging and very helpful remarks. With the above revisions to the manuscript, we hope that the referee would be willing to recommend the acceptance of our work by Nature Communications.

Reviewer #6 (Remarks to the Author):

In the manuscript by Wong et al., the authors introduce a rigorous theoretical formalism aimed at demonstrating that the rate of quantum electrodynamical (QED) interactions, which lead to photon emission, can be controlled via electron wave shaping. In particular, they show that free-electron wave shaping can be used to tailor both the spatial and the spectral distribution of the emitted photons. To demonstrate this, the authors apply the results of their theory to the illustrative case of Bremsstrahlung. Spatial tailoring is theoretically predicted by analyzing the interaction between an electron in a superposition of multiple (i.e. two) momenta and a Carbon atom. Such an interaction leads to an enhanced directionality of the emitted photons. Similarly, spectral tailoring is theoretically verified by analyzing the interaction between an electron (in a superposition of multiple, i.e. two, momenta) and an undulator system. Such a process leads to an enhanced monochromaticity and directionality of the emitted photons.

The authors claim to have demonstrated for the very first time that the properties of the photons emitted in QED processes can be tailored through shaping of the electron wave packet. This is somehow unprecedented and of high interest and impact for several applications, such as energy dispersive X-ray spectroscopy (EDS) and electron energy loss spectroscopy (EELS). The applicability of the proposed theory is indeed testified by the fact that the authors consider the potential practical example of Bremsstrahlung, which is routinely employed in state-of-the-art X-ray sources for medical imaging, security scanning, materials analysis, and astrophysics. The new approach introduced in this manuscript can thus open novel avenues to technologies that are based on free-electron wave-shaping (for example, medical imaging, industrial quality inspection, and security scanning).

Authors' reply: We thank the referee for highlighting the novelty and high interest of our manuscript.

As can be inferred from the referees' comments, as well as from the authors' response, the previous version of the manuscript was significantly different from the current one. In particular, in a first stage, the authors made use of a semiclassical approach based on solving the Schrödinger equation for the electron wavefunction in order to assert their claim. However, some reviewers (especially, referee #2) questioned such a semiclassical treatment, since it did not (and could not)

provide a rigorous and solid theoretical demonstration of all their findings. The revised manuscript now presents a complete and more meticulous analysis that is entirely based on the QED theory and that provides strong theoretical evidence of their claims. I believe that the new version of the manuscript definitely raises the level of the entire work.

Authors' reply: We are grateful for the referee's positive feedback and appreciation of the complete QED theory and for emphasizing its value.

Yet, despite its excellence and potential high impact, this manuscript still presents what we believe is a relevant issue, i.e. the lack of an experimental observation supporting the new developed theory, as already pointed out by all the other reviewers (especially by referee #1). A valid experiment accompanying and evidencing the theory is, we believe, typically expected in journal such as Nature Communications. The authors mention throughout the text some practical methods currently available to generate the required electron beams. In particular, they stress the fact that their theory could be directly applied in a simple fashion to electron energies provided by sources commonly feeding scanning electron microscopes (SEMs) and transmission electron microscopes (TEMs). However, such asserts cannot be in principle considered, in our opinion, either as an exhaustive argument or a convincing experimental evidence. Furthermore, the authors do not explain or suggest how to achieve wave shaping in a practical scenario, even though they mention some experiments in which their outcomes could be experimentally verified. Overall, the lack of an experimental evidence makes the manuscript difficult to assess.

Authors' reply: We thank the referee for raising this point. The experimental prospects of this theory are indeed exciting and highly promising. The theory of quantum electrodynamics (QED), from which our theory is derived *ab initio*, has repeatedly stood the test of experimental verification, from Compton scattering in 1923 to undulator radiation in modern state-of-the-art synchrotrons. However, it was not until recent times that precise methods of spatiotemporal electron control started rapidly attracting interest, providing the impetus for us to discover this unexplored side of QED: quantum enhancement through electron waveshaping.

To fully address the referee’s concern, we have expounded on methods to achieve electron waveshaping relevant to realizing these quantum enhancement experiments. In particular, the splitting of an electron into two interfering beams – which we use in both our examples – can be achieved using an electron biprism or single crystal thin films, allowing our examples to be studied using available technology. In our revised submission, we present specific examples of experiments that demonstrated such electron waveshaping in both TEMs and SEMs.

The most important revision of the manuscript is in detailing proposals for experimentally demonstrating quantum control of bremsstrahlung processes via electron waveshaping. This technical proposal is explained in the newly added Supplementary Information Section S2, and also discussed in the main text. Our calculations show that using table-top electron sources and realistic electron deflection angles from a biprism, we are able to obtain substantial photon emission rates in atomic bremsstrahlung, as well as significant measurable changes in emission characteristics moving from the unshaped to the shaped electron wavepacket cases. Our calculations reveal the specific conditions needed to access this mechanism, and indicate that such an experimental demonstration is possible with modern experimental equipment. These results thus make an even stronger case for the impact and timeliness of our theory.

In conclusion, the manuscript introduces a novel methodology that could be of high interest, potentially bringing about a great impact for scientists working in the specific fields of electron wave-shaping applications, as well as for related technologies (medical imaging, industrial quality inspection, and security scanning). However, as an experimental investigation supporting the theoretical analysis is missing, the manuscript would probably be of higher value and impact with some experimental evidence or perhaps more suitable for a journal like Phys Rev Letter. Nevertheless, I am not contrary to the publication in Nature Communications, if the Editor is ready to take some risk in terms of future feasibility, as the theoretical approach is very interesting per se.

Authors’ reply: We thank the referee for the glowing remarks on our work, and also for the extremely helpful comments. The remaining concern on “risk in terms of future feasibility”

is directly addressed in our improved manuscript through newly added experimental proposals showing that the required experimental parameters are well within reach of the technology available today. These experimental designs are backed up by quantitative analyses that confirm their feasibility under realistic experimental conditions. With these revisions, we hope that the referee would be able to support our work for publication in Nature Communications.

Reviewers' Comments:

Reviewer #5:

Remarks to the Author:

My questions have been well answered and the manuscript could be accepted by Nature Communication.

Reviewer #6:

Remarks to the Author:

The authors responded to the reviewers' comments and suggested possible experimental outcomes. I am therefore comfortable in recommending publication.

Response to Reviewers

Reviewer #5 (Remarks to the Author):

My questions have been well answered and the manuscript could be accepted by Nature Communication.

We thank the reviewer for the positive recommendation on our work.

Reviewer #6 (Remarks to the Author):

The authors responded to the reviewers' comments and suggested possible experimental outcomes. I am therefore comfortable in recommending publication.

We thank the reviewer for supporting our work for publication in Nature Communications.